# Migraine, Tension-Type Headache and Parkinson’s Disease: A Systematic Review and Meta-Analysis

**DOI:** 10.3390/medicina58111684

**Published:** 2022-11-20

**Authors:** Efthalia Angelopoulou, Andreas Nikolaos Papadopoulos, Nikolaos Spantideas, Anastasia Bougea

**Affiliations:** 1Department of Neurology, Medical School, Eginition Hospital, National and Kapodistrian University of Athens, 11527 Athens, Greece; 2Department of Hygiene, Epidemiology and Medical Statistics, School of Medicine, National and Kapodistrian University of Athens, 11527 Athens, Greece

**Keywords:** Parkinson’s disease (PD), migraine, tension-type headache (TTH), non-motor symptoms (NMS), prevalence, unified Parkinson’s disease rating scale (UPDRS)

## Abstract

*Background and Objectives*: The relationship between migraine and tension-type headache (TTH) with Parkinson’s disease (PD) is controversial, while a common pathophysiological link remains obscure. The aim of this systematic review is to investigate the association between PD, migraine and TTH. *Materials and Methods*: Following PRISMA, we searched MEDLINE, WebofScience, Scopus, CINAHL, Cochrane Library and ClinicalTrials.gov up to 1 July 2022 for observational studies examining the prevalence and/or associations of PD with migraine and TTH. We pooled proportions, standardized mean differences (SMD) and odds ratios (OR) with random effects models. The risk of bias was assessed with the Newcastle-Ottawa scale (PROSPERO CRD42021273238). *Results*: Out of 1031 screened studies, 12 were finally included in our review (median quality score 6/9). The prevalence of any headache among PD patients was estimated at 49.1% (760 PD patients; 95% CI 24.8–73.6), migraine prevalence at 17.2% (1242 PD patients; 95% CI 9.9–25.9), while 61.5% (316 PD patients; 95% CI 52.6–70.1) of PD patients with migraine reported headache improvement after PD onset. Overall, migraine was not associated with PD (302,165 individuals; OR_pooled_ = 1.11; 95% CI 0.72–1.72).However, cohort studies demonstrated a positive association of PD among lifetime migraineurs (143,583 individuals; OR_pooled_ = 1.54, 95% CI 1.28–1.84), while studies on 12-month migraine prevalence yielded an inverse association (5195 individuals; OR_pooled_ = 0.64, 95% CI 0.43–0.97). Similar findings were reported by 3 studies with data on the TTH-PD relationship (high prevalence, positive association when examined prospectively and an inverse relationship on 12-month prevalence). These data were not quantitatively synthesized due to methodological differences among the studies. Finally, PD patients suffering from any headache had a lower motor unified Parkinson’s disease rating scale (UPDRS) score (503 PD patients; SMD −0.39; 95% CI −0.57 to −0.21) compared to PD patients not reporting headache. There is an unclear association of headaches in genetic PD cohorts. *Conclusions*: Observational data suggest that migraine and TTH could be linked to PD, but the current literature is conflicting.

## 1. Introduction

Although Parkinson’s disease (PD) is classified as a motor neurodegenerative disorder, non-motor symptoms (NMSs)are common, such as sleep disorders, depression, cognitive deficits and pain [1]. Among PD patients, different pain phenotypes and/or classification shave been described: PD-related (including fluctuation-related, dyskinesia -related and central pain) vs. non-PD-related pain; primary vs. secondary (musculoskeletal system) pain; nociceptive vs. neuropathic; chronic vs. acute; and pain according to different body parts [2,3,4,5]. However, the exact pain etiology in PD remains unclear; as a consequence, therapeutic options are often ineffective [6].

Primary headaches (i.e., migraine and tension-type headache (TTH)) are commonpain disorders with several comorbidities, such as cardiovascular disease, stroke, depression and various dementias, and are accompanied by a substantial socioeconomic impact [7,8].Over the previous two decades, several studies have suggested a link between primary headaches and PD [9,10,11], while others have produced conflicting results [12,13]. Barbanti et al. identified a lifetime migraine prevalence of 27.8% and a current migraine prevalence of 13.1% in a population of PD patients [14]. Moreover, a cohort study showed that subjects with a midlife history of headache and, particularly, those with migraine with aura had an increased likelihood of PD [9].By contrast, Nunes et al. [15] demonstrated that PD patients had a lower lifetime prevalence of headache than controls. Previous studies have focused on the relationship between migraine and PD, and a prior validation study reported that 80% of individuals with non-migrainous headache might have TTH [16]. However, the supposed overlapping PD and headache pathophysiological mechanisms remain obscure [17]. Despite the high prevalence of migraine across various PD populations [14,15], headache remains an underestimated and undertreated NMS contributing to a poor quality of life among PD patients [14].

A clarification of the examined relationship will provide valuable knowledge to neurologists in determining the modifiable causes of poor outcomes.

Here, we systematically reviewed data on the association between migraine, TTH and PD, focusing on epidemiology, clinical characteristics and pathophysiology. We sought to test the hypothesis that patients with migraine or TTH have a higher risk for PD in later life in comparison to individuals not suffering from primary headaches.

## 2. Materials and Methods

This systematic review and meta-analysis was conducted according the updated Preferred Reporting Items for Systematic reviews and Meta-Analyses (PRISMA) [18]. The protocol was registered with PROSPERO (ID CRD42021273238).

### 2.1. Search Strategy

Two independent reviewers (EA and AB) carried out a literature search in PubMed, Web of Science, Scopus, CINAHL, Cochrane Central Register of Controlled Trials and ClinicalTrials.gov (from inception to 1 July 2022) to identify studies examining the relationship between PD and migraine or TTH. The search was restricted to the English language and only studies with primary data were included. The Mesh terms used were: “Parkinson’s disease” AND (“migraine” OR “tension-type headache”).The grey literature searches included Open Grey and conference posters. References of included studies were also hand-searched for eligibility (snowball procedure).

### 2.2. Study Selection, Inclusion and Exclusion Criteria

The studies were assessed following a 2-level screening process. Level 1 entailed title and abstract screening of the records retrieved. The full text of all citations deemed potentially eligible was retrieved and screened for final eligibility. Both levels were conducted by2 independent reviewers (EA and AB), and any discrepancies were resolved by team consensus involving all authors. Screening was conducted based on predefined eligibility criteria, as described below and in our pre-published protocol.

Inclusion criteria included: (a). Adult PD participants with migraine or TTH or individuals suffering from migraine or TTH who were prospectively followed for incident PD onset; (b). Any prospective or retrospective observational study [19]; (c). Cross-sectional studies for prevalence outcomes (migraine and TTH among PD patients); and (d). Studies reporting an association estimate with corresponding 95% confidence intervals (CIs) or providing appropriate data, which can be used to calculate a crude estimate.

Exclusion criteria included: (a). Irrelevant studies; (b). Other primary or secondary headaches attributed to a disorder other than PD or tomuscle contraction; (c). Studies on atypical parkinsonism or other movement disorders; (d). Articles published in a language other than English; (e). Animal studies; (f). Study protocols, letters/commentaries/communications, case reports or series, editorials or conference abstracts; and g. Literature, systematic or umbrella reviews and meta-analyses.

### 2.3. Data Extraction and Quality Assessment

Two authors (EA and AB) manually extracted data from full texts and a third independent reviewer (ANP) further verified the data. From each eligible study we extracted the study design, sample size, headache type, PD and headache diagnostic criteria, the effect of any medication received (effect of antiparkinsonian therapy on intensity, frequency or duration of headache and antimigraine therapy on PD symptoms), tools for assessing headaches and PD motor/NMS, as well as proportions and association estimates for our meta-analysis outcomes.

Having retrieved the full text of articles that met the inclusion criteria, two authors (EA and AB) independently assessed the methodological quality of the studies. Case–control and cohort studies were evaluated according to the Newcastle–Ottawa scale (NOS), while a modified NOS version was used for cross-sectional studies. The NOS contains nine items, categorized in 3 dimensions: (a). Subject selection (score range, 0–4); (b). Comparability of subjects (score range, 0–2); and (c). Clinical outcomes for cohort studies or exposure for case–control studies (score range, 0–3). A star system was used to allow a semi-quantitative assessment of study quality, such that the highest quality studies are awarded a maximum of one star for each item with the exception of the item related to comparability that allows the assignment of two stars. NOS scores range from 0 to 9 (good quality: ≥7, moderate quality: 3–6 and poor quality: ≤2). For cross-sectional studies, we used the modified 7-item NOS scale (good study quality: ≥6, moderate quality: 3–5 and poor quality: ≤2). Discrepancies were resolved by team discussion.

### 2.4. Statistical Analysis

From each included study, we extracted all relevant association estimates along with their corresponding 95% CI. We carried out 3 approaches: (a). Meta-analysis of proportions; (b). Meta-analysis of standardized mean differences (SMD); and (c). Meta-analysis of odds ratios (OR). In the case of proportions, we sought to stabilize the variance and achieve approximate normality by utilizing the Freeman–Tukey (FT) double arcsine transformation. This method allows admissibility of all studies (on the contrary, the logit transformation may exclude studies with proportions near 0 or 100%), while the pooled CI always lie within the desired range of 0–100% [20,21]. Harmonic mean was used in the back-transformation formula [21]. For each meta-analysis of proportions, we also reported the “Raw proportion (%)”, which is derived by simple weighted division. Results were reported in the following format: number of studies pooled; total PD patients; raw proportion; pooled proportion; 95% CI; and I^2^. In the case of SMD, taking into account that sample sizes were relatively small, we used the method of Hedge’s in order to take advantage of the correction factor and report more conservative results. Finally, out of the 7 studies reporting association ratios, 5 presented OR while 2 presented hazard ratios (HR). Because the outcome (incident PD) was relatively rare in both of these cohorts (<<10%) [10,22], we considered HR and OR to be comparable and applied no transformation [23]. In all cases, we carried out random effects meta-analyses with the estimate of tau^2^ being inferred from the iterative restricted maximum likelihood (REML) method. Where study samples were sufficient, we also performed subgroup and/or sensitivity analyses by study design, as well as headache duration (all studies, headache prevalence at any time during lifetime (“lifetime headache prevalence”) or headache prevalence during the last 12 months (“12-month headache prevalence”)).

Statistical heterogeneity was quantified via I^2^ (significance threshold of *p* < 0.10), computed by Cochran’s Q and was categorized as not important (0–40%), moderate (40–60%), substantial (50–90%) or considerable (90–100%) [24].

The effect of potential publication bias (small-study effects) was explored using the Egger’s test(significance threshold of 0.10) [25,26].

Statistical significance was set at a two-sided *p* < 0.05. All analyses were conducted via the STATA software version 16.1 (Stata Corporation, College Station, TX, USA).

## 3. Results

### 3.1. Description of Included Studies

From a total of 1031 identified articles, 391 were identified as duplicates and a further 614 were excluded following title and abstract screening. After full-text screening, 28 articles were additionally excluded because they did not meet the inclusion criteria. Finally, we included 12 eligible studies and pooled results from 11 of those studies. The selection process is illustrated in a PRISMA flowchart (Figure 1, Appendix A).

### 3.2. Study Characteristics

The clinical characteristics and main outcomes of the included studies are summarized in Table 1. Four out of 12 eligible studies employed a cross-sectional, case–control or cohort design (mean follow-up time: 5.25 years). The remaining 3 studies utilized a cross-sectional design for the prevalence outcomes while also reporting association estimates via a case-control design. None of the studies were an RCT. Two studies had sample sizes over 15,000 [22,27], while the other 10 had sample sizes below 1000 [9,10,11,14,15,28,29,30,31,32]. Seven out of the 12 studies had a moderate quality (NOS score of 6) (Appendix A).

### 3.3. Headache Prevalence in PD Patients

The overall pooled prevalence of any headache (including migraine and TTH) among PD patients was 49.1% (5 studies; 760 PD patients; 40.9%; 49.1% with 95% CI 24.8–73.6; and I^2^ = 97.3%, Figure 2). The 12-month prevalence of migraine among PD patients was calculated at 13.1% (3 studies; 771 PD patients; 11.2%; 13.1% with 95% CI 3.8–26.2; and I^2^ = 94.1%), while the lifetime prevalence was expectedly higher at 15.7% (5 studies; 1144 PD patients; 14.5%; 15.7% with 95% CI 7.9–25.2; and I^2^ = 92.3%, Figure 2). Only very few PD patients had received a diagnosis of migraine with aura (3 studies; 719 PD patients; 2.6%; 0.2% with 95% CI 0.0–8.6; and I^2^ = 93.5%).

### 3.4. Association between PD and Migraine

Overall, regarding the relationship between PD and migraine, no statistically significant association was found (7 studies; 302,165 individuals; OR_pooled_ = 1.11, 95% CI 0.72–1.72; and I^2^ = 92.0%; Figure 3).

This was also the case when pooling studies specifically on migraine with or without aura (Figure 4).

However, when restricting our analysis to the 2 cohort studies on lifetime migraine prevalence, the result achieves significance and translates to an increased risk of PD among lifetime migraineurs (2 studies; 143,583 individuals; OR_pooled_ = 1.54, 95% CI 1.28–1.84; and I^2^ = 0%, Figure 4). Notably, case-control studies failed to show a statistically significant association, possibly due to considerable between-study heterogeneity (Figure 3). On the contrary, in studies focusing only on a 12-month migraine prevalence, an inverse association with PD was found (3 studies; 5195 individuals; OR_pooled_ = 0.64, 95% CI 0.43–0.97; and I^2^ = 0%, Figure 4). These contrasting findings could be partially explained by the potentially favorable effect of PD on the course of migraine if seen together with the result regarding the proportion of PD patients reporting migraine improvement after PD onset (3 studies; 129 PD patients; 61.2%; 61.5%, 95% CI 52.6–70.1; and I^2^ = 0%, Figure 2).

### 3.5. Publication Bias

The Egger’s test detected publication bias on the analyses associating migraine with PD (*p* = 0.04 for all studies and *p* = 0.04 for lifetime migraine). When performing subgroup analyses according to the study design, this effect seemed to resolve, implying that heterogeneity could be the cause of asymmetry. However, the possibility of residual small-study effects cannot be excluded due to the very few studies pooled and the inherent loss of power associated with Egger’s test when testing <10 pooled studies.

### 3.6. Differences in Motor Symptoms between PD Patients with or without Headache

When pooling SMDs, PD patients suffering from headache of any duration had on average a lower score in the motor section (III) of the UPDRS scale compared to PD patients not reporting headache (3 studies; 503 PD patients; SMD −0.39; 95% CI −0.57 to −0.21; and I^2^ = 0%, Appendix A). Restricting this analysis to the 2 studies on PD patients with migraine, the result remained stable in direction and magnitude, albeit losing significance (Appendix A).

### 3.7. Association between PD and TTH

Only 3 studies [15,22,30] examined the prevalence and/or associations between TTH and PD with conflicting results. We were unable to pool this data due to the vastly different methodological designs of the respective studies. Yang et al. suggest that TTH suffering individuals are at an increased risk of incident PD later in life. This is in line with the findings from a small PD patient sample by Sampaio Rocha-Filho et al. [30], where TTH prevalence among PD individuals was high (67.4%). On the contrary, Nunes et al. [15] suggest that PD is associated with a lower TTH 12-month prevalence. These results are directionally similar with the ones pertaining to the PD-migraine association.

### 3.8. Genetics and Family History of Migraines in PD Patients

Whether genetics and a family history of migraines in PD patients influences the course of PD is not clear, as relevant data are reported in only 3 studies [13,14,22]. The course of migraine after the onset of PD in patients without a family history of migraine differed to that of those with a family history [22]. PD onset seemed to alter the course of migraine among migraineurs without a family history of migraine but not among those with a self-reported family history. [13]. However, Barbanti et al. [14] indirectly refute this finding, as almost half of PD patients without a family history of migraine had current migraine. Moreover, the lower prevalence of family history of migraine among PD migraineurs could not be explained.

### 3.9. Therapy Effect

Five studies described the effects of levodopa and dopamine agonist (DA) drugs on headache course. Mecoet al. [11] did not reveal any significant differences between PD groups with and without headache [11]. Van Hilten et al. [13] concluded that the dose of levodopa was not related to the course of migraine attacks in a small sample of 5 PD patients [13]. There was also no significant difference in a levodopa equivalent daily dose (LEDD), irrespective of headache comorbidities [32]. By contrast, Barbanti et al. [14] suggested that the beneficial prophylactic effects of dopaminergic therapy would better explain the improvement in migraine in some PD patients than the remission of the disease in others [14]. Cubo et al. [29] found that lower doses of PD medication reached better ON scores in PD patients with migraines compared with those without migraine [29].

## 4. Discussion

The present systematic review and meta-analysis suggests a potential association between migraine, TTH and PD. The main results are summarized as follows: (a). Migraine and ΤΤH are relatively common among PD patients and may be associated with PD; (b). A lower score in the motor section of the UPDRS scale was found in PD patients with headache compared to PD patients not reporting headache; (c). The majority of PD patients reported migraine improvement after PD onset; (d). There is an unclear association of headaches in genetic PD cohorts; and e. A complex response of headache to dopaminergic drugs is described in the literature.

We found that in studies examining lifetime migraine prevalence, the results are in favor of a positive association between migraine and PD. By contrast, studies focusing on12-month prevalence found an inverse association between migraine and PD. This could be partially explained by the potentially favorable effect of PD on migraine headache course and may also explain why the analysis pooling all studies yields an overall non-significant result. Case-control studies reported a very wide range of prevalence of migraines in PD patients from 6.7 to 40.8% [15,29,32]. The vast differences in prevalence could be explained by different methodological approaches, such as questionnaires and face-to-face interviews, leading to an inconsistent reporting of symptoms among PD patients and inaccurate data collection on the part of neurologists in the setting of movement outpatient clinic.

An interesting result was the lower motor UPDRS score in PD patients with headache compared to PD patients without headache. However, this finding is difficult to interpret because of the heterogeneity of the UPDRS versions across the different studies [29,30,32]. Patients with PD usually experience a worsening of motor symptoms during the OFF state. However, the evaluation of headaches was not made specifically during the OFF state, which might explain the motor aggravation of PD patients [15].

Dopamine may be implicated in modulating pain processing in different areas of the brain, such as the spinal cord, thalamus, periaqueductal gray matter, basal ganglia and cingulate gyrus. The mesolimbic and nigrostriatal pathways originate in the substantia nigra pars compacta and project to the dorsal striatum [33]. These pathways are linked with various NMS in patients with PD, such as pain. Moreover, the pain tolerance threshold in patients with PD tends to be lower than that exhibited by healthy individuals. This could be ascribed to damage in the descending inhibition of nociception from supraspinal structures [34]. Inadequate descending pain modulation has also been reported to be instrumental in the pathophysiology of chronic TTH [35]. Consequently, impaired dopaminergic function and deficient descending pain inhibition from supraspinal sites have emerged as potential shared mechanisms between PD and TTH. Serotonergic dysfunction has also been implicated in the pathophysiology of PD [36], with serotonergic cell loss described to appear even before nigrostriatal dopaminergic degeneration in PD [37]. In fact, previous animal studies demonstrate that serotonin release is blocked following the blockade of postsynaptic D2 receptors and that serotonin release in the forebrain entails an intact local dopaminergic neurotransmission [38]. Furthermore, serotonergic deficiency assists in the impaired descending inhibition of pain and the central sensitization of nociceptive pathways, which are thought to be implicated in the pathophysiology of TTH [39]. In light of the overlap between these processes, TTH may have a potential pathological link with PD.

Common mechanisms linking migraine with PD have also been suggested, namely, the dysfunction of the serotonergic and dopaminergic system. Serotonin depletion has been associated with increased cortical neuron sensitivity and vasoconstriction induced by cortical spreading depression in an animal model of migraine [40,41]. Except for PD, dopaminergic dysfunction has also been hypothesized as a causal factor in migraine pathogenesis, as excessive yawning, nausea and vomiting are common prodromal and accompanying symptoms of migraine. This could be related to the stimulation ofDA2 receptors, which are abundant in migraineurs with aura [42]. Iron, an essential nutrient for DA regulation, was elevated in the substantia nigra of PD patients, leading to toxic oxygen radicals and DA cell death [43]. Furthermore, increased iron accumulation in deep brain nuclei in migraineurs may be suggestive of impaired antinociceptive pathways [44]. So far, whether “Iron accumulation could be an indice of tissue damage due to other processes or a reversible epiphenomenon” cannot be answered [43]. If the increased iron deposition observed in migraineurs in these few studies reflects (directly or indirectly) neuronal damage, this would be consistent with late-life PD symptoms. By contrast, studies included in the present review did not investigate the relationship between iron accumulation and migraine.

Another interesting finding that emerges from this review is the amelioration or remission of migraine after PD onset [14,31,32]. Clinical and pharmacologic data suggest that there is DA receptor hypersensitivity in migraine. The use of dopaminergic drugs in migraineurs can precipitate migraine attacks, including aura and pain. Moreover, there is dopamine receptor hypersensitivity in migraineurs, as demonstrated by the induction of yawning, nausea, vomiting, hypotension and other symptoms of a migraine attack by dopaminergic agonists at doses that do not affect non-migraineurs [17]. In migraineurs, these effects can occur at lower-than-average doses [45,46,47], and apomorphine can, itself, trigger migraine [48]. Finally, DA antagonists such as promethazine and prochlorperazine were seen to alleviate the gastrointestinal symptoms of migraine attacks [49], but also provoke extrapyramidal symptoms. Substantia nigra possesses a high concentration of serotonin 1—B, D and F—receptors that are the target of sumatriptan, an antimigraine agent, and this could alter tolerability of sumatriptan in PD. These data suggest a complex relationship between migraine and PD because DA plays simultaneously therapeutic and pathogenic roles in migraine. However, the improvement of migraine in patients with PD has also been attributed to aging via the degeneration of pain regulatory systems, regardless of PD [31]. This may be due to decreased dopamine sensitivity with aging, without serotonergic involvement. However, it is difficult to determine whether headache improvement is due to the degeneration of the brainstem regions involved in headache transmission or to the effects of dopamine agonists.

This review has several limitations. The included studies have methodological issues such as the retrospective nature, the self-reported outcomes, the lack of follow-up (except in the few cohort studies), the absence of comparator groups when presenting proportion estimates [28,30] and no uniformity across the studies with different UPDRS versions [29,30].The majority of the studies fulfilled standardized diagnostic criteria for headaches [13,14,15,22,29,31], while in few, the diagnosis of headache was based on diagnostic codes. There was no additional clinical information on TTH, migraine, PD stage or comorbidities [10,22]. Furthermore, surveillance bias might have exaggerated the association between migraine [9]. The majority of included studies in the present review were not designed to account for confounding factors, and most analyses were crude. Indeed, many women with PD in their sixth decade go through menopause, and it is well known that the hormonal transition during menopause has a significant association with migraine. Additionally, many PD patients with orthostatic hypotension who are on symptomatic treatment with midodrine also have supine hypertension, which frequently causes headaches (often disguised as TTH). Although our search included six databases and the reference lists of all articles that met the inclusion criteria, the potential of publication bias cannot be completely excluded. However, our search strategy had the advantage of being rigorous, allowing an exhaustive literature review. Other limitations consist of the high heterogeneity in the estimation of the pooled prevalence estimates and ORs, the inclusion of studies with small sample sizes and the inability to perform meta-regression analyses (due to few studies). The majority of studies available for analysis were of cross-sectional design, and selection bias could not be ruled out; a strength of our study lies in the quality assessment of included studies. In particular, according to NOS scores, the majority of the studies included had a score of ≥6, suggesting a moderate quality.

### Recommendations for Future Research

We recommend that future studies include longitudinal design, larger samples and a longer follow-up. Preferably, the comparisons of PD groups with genetic mutations, with device-aided therapies in conjunction with oral PD medications vs. a pharmacotherapy-only control group would clarify the effects of PD therapy on headaches. It is also crucial to examine the effects of antimigraine drugs on PD symptoms. D2 antagonist antiemetics should be avoided for migraine treatment among PD patients. There is an undeniable need to apply validated diagnostic criteria and standardized tests to evaluate headaches in PD patients, including biomarkers (blood hormones, salivary cytokines and neuroimaging) [50,51]. A careful headache history and diagnostic work-up is essential to rate underlying pathophysiology and to establish treatment recommendations for PD patients. Studies should also investigate whether PD patients might benefit from treatment targeting migraine and alleviating motor symptoms (stiffness).

## 5. Conclusions

This systematic review revealed that migraine and TTH commonly co-exist with PD. However, the potentially higher risk of incident PD among migraineurs in cohort studies was not verified (and even reversed) in sensitivity analyses. Future research should aim to elucidate the abovementioned associations and their common mechanisms via large well-designed prospective multicenter studies.

## Figures and Tables

**Figure 1 medicina-58-01684-f001:**
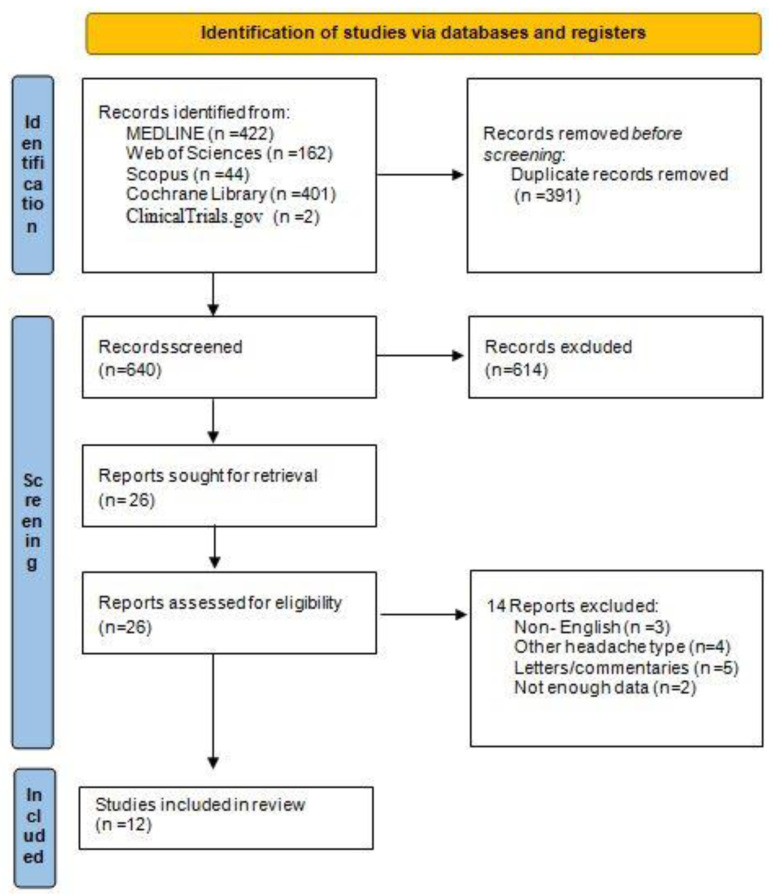
PRISMA 2020 flowchart.

**Figure 2 medicina-58-01684-f002:**
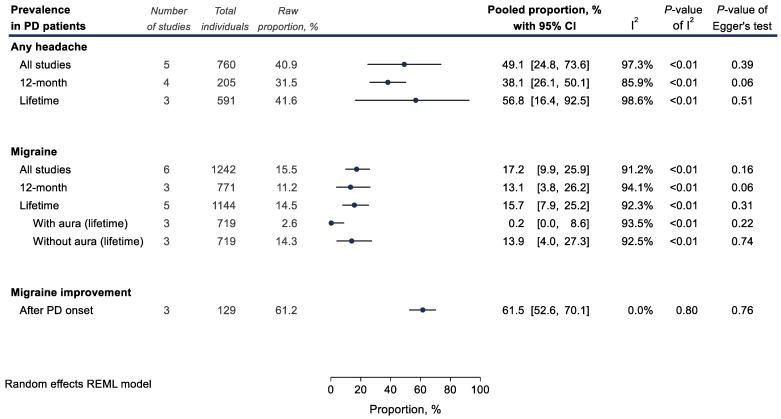
Meta-analyses of proportions examining headache prevalence (%) among Parkinson’s disease (PD) patients. Results of each individual random effects meta-analysis are depicted as black data markers; 95% confidence intervals (CI) are indicated by the black error bars. I^2^ reflects each meta-analysis heterogeneity.

**Figure 3 medicina-58-01684-f003:**
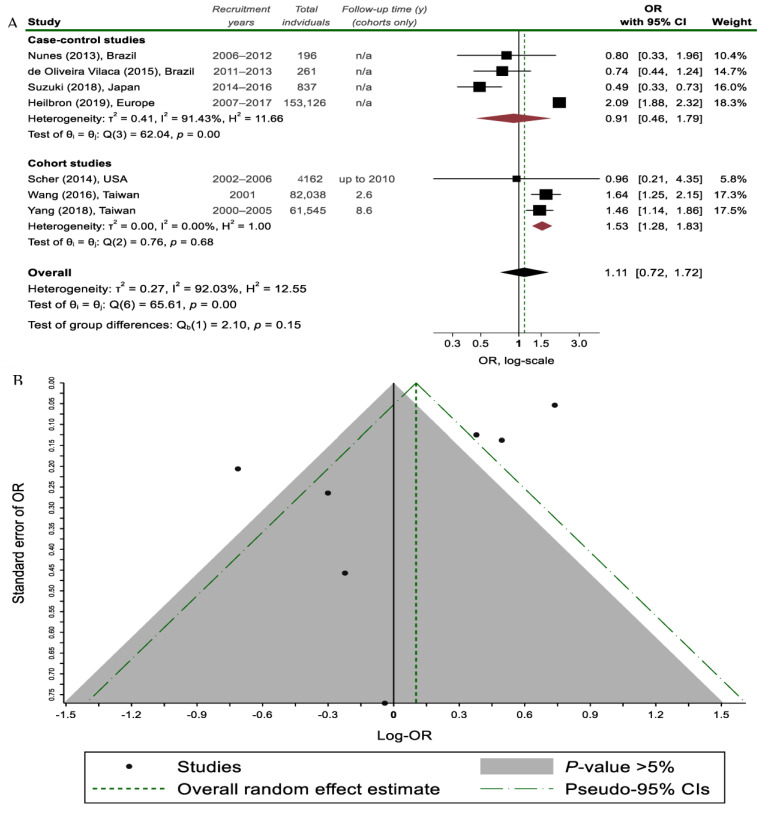
(**A**) Random effects meta-analysis on the association between migraine headaches and Parkinson’s disease (PD) status stratified by study design; (**B**) Funnel plot, where each study is depicted as a dot. The light grey shaded area contains studies with non-significant results (*p* > 0.05) [7,9,15,22,27,31,32].

**Figure 4 medicina-58-01684-f004:**
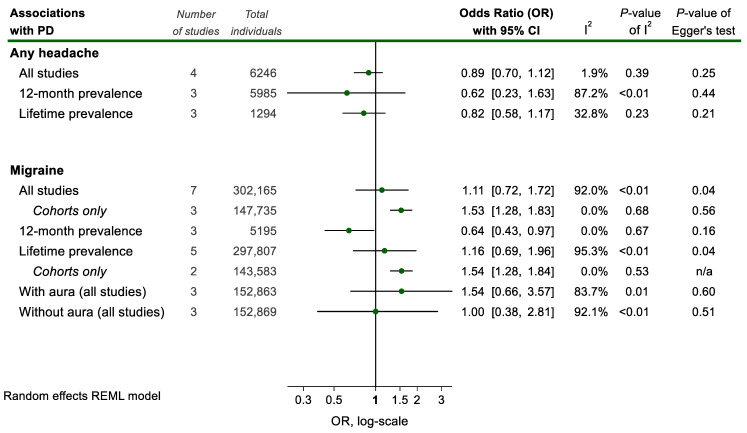
Meta-analyses of associations between headache and Parkinson Disease (PD) status. Results of each individual random effects meta-analysis (odds ratios (OR)) are depicted as green data markers; 95% confidence intervals (CI) are indicated by the black error bars. I^2^ reflects each meta-analysis heterogeneity.

**Table 1 medicina-58-01684-t001:** Summary of main characteristics and outcomes of patients with PD and migraine or TTH.

Author, Year	Study Design, N Total Sample, Mean Age, Sex (m/f), Mean Follow-Up Time	Headache Prevalence * %	Standardized Criteria for PD Diagnosis	Standardized Criteria for Headaches Diagnosis	Headache Assessment	Measures for Motor/Non-Motor PD Symptoms	Medication Effect	Main Outcomes
Indo, 1983 [28]	Cross-sectional, *n* = 25 PD (11.25% headache with pulsation characteristics) Mean age:34.4 y, 8 m/17 f	35.2%	NA	NA	Interview	NA	NA	No correlation between nuchal rigidity and occipital pain in PD patients with headache.
Meco, 1988 [11]	Case–control, *n* = 17 9 PD and H, mean age: 67 y,5 m/4 f 8 PD and nonH, mean age: 68 y, 4 m/4 f	NA	NA	NA	NA	Motor WRS, CURS, HY Non-motor HRS, Zung RS, GDS, STAI X-1 MMSE, HAD	No effect of antiparkinsonian therapy on headaches	Higher muscular activity, anxiety and depression in PD patients with headaches vs. C.
Barbanti, 2000 [14]	Population cross-sectional, *n* = 132 4 PD and M, 62 PD and MWA, mean age: 63.6 y, 20 m/46 f 5 HC and M, 61 HC and MWA, mean age: 61.5 y, 20 m/46 f	Lifetime M prevalence:27.8% Current M prevalence:13.1%	UK PD Society Brain Bank	ICHD-I	Structured questionnaire on M family history	Motor HY, UPDRS Non-motor NA	Dopaminergic therapy improves M	Lower M frequency of PD patients vs. C. After PD onset, 2/3 of PD patients reported an improvement in or remission of migraine.
Cubo, 2004 [29]	Cross-sectional, single-blinded clinical trial, *n* = 28 10 PD and M, mean age: 63.6 y 18 PD and nonM, mean age: 61.5 y	M prevalence: 8%	NA	ICHD-I		Motor ON/OFF stage HY, ON/OFF stage UPDRS III, IV (dyskinesias) Non-motor UPDRS IV (hallucinations)	Lower dopaminergic dose for PD and M vs. PD and nonM	Improved ON motor state of PD and M vs. PD and nonM patients with the same LEDD.
Nunes, 2013 [15]	Cross-sectional, *n* = 98 26 PD and M, mean age: 57.4 y, 18 m/22 f 12 PD and TTH, mean age: 62.8 y, 39 m/19 f 31 HC and M, mean age: NA, sex: NA 33 HC and TTH, mean age: NA, sex: NA	M: 26.5% TTH: 12.2%	UK PD Society Brain Bank	ICHD-II	Standardized questionnaire	Motor UPDRS III Non-motor NA	NA	Lower prevalence of primary headache in PD patients vs. HC in previous year. The predominant side of headache was ipsilateral to the side of initial PD motor symptom.
Scher, 2014 [9]	Population-based, prospective AGES cohort, *n* = 65 (tot.no PD with headache and C) NMH = 10, mean age: NA, sex: NA MA = 10, mean age: NA, sex: NA MWA = 2, mean age: NA, sex: NA C = 43, mean age: NA, sex: NA Follow-up:4 y	NMH: 2.5% MWA:3.4% MA: 4.9%	NA	ICHD-I	Interview	Motor 6 screening questions for parkinsonism 6 m walking speed test Non-motor NA	NA	PD diagnosis was more likely among subjects with MA (OR_MA_ 5 2.5 (95% CI 1.2–5.2)) later in their life.
DeOliveira Vilaça, 2015 [31]	Cross-sectional, *n* = 58 21 PD and M, mean 65 y, sex: NA 37 C and M, mean age: NA, sex: NA	M:19%	UK PD Society Brain Bank	ICHD-II	Questionnaire on M features	NA	NA	The recovery of M in PD patients may be attributed to senescence.
Wang, 2016 [10]	Population-based, prospective cohort, 148 PD and M, mean age: NA, sex: NA 101 PD nonM, mean age: NA, sex: NA Follow-up:39.5 months	M:0.36%	ICD-9-CM code 332.0	ICD-9-CM code 346	NA	NA	NA	M group was at higher risk of developing PD (HR = 1.64 95% CI: 1.25–2.14, *p* = 0.0004) compared with the nonMgroup.
Yang, 2018 [22]	Population-based, prospective cohort, *n* = 61,545 320 PD and TTH, mean age: NA, sex: NA 720 PD and nonTTH, mean age: NA, sex: 76 PD and M, mean age: NA, sex: NA Follow-up:8.55 y (TTH), 8.63 (nonTTH)	NA	ICD-9-CM code 332	ICHD-II	NA	NA	NA	Overall PD incidence = 3.01 (TTH), 1.68 (nonTTH) higher PD risk in TTH vs. nonTTH (adjusted HR = 1.37, 95% CI = 1.19–1.57) even after adjustment for sex, age and comorbidities.
Suzuki, 2018 [32]	Multicentercross-sectionalcase–control, *n*= 837 29 PD and M, mean age: 70 y, 10 m/19 f 139 PD and nonM, mean age:68.4 y, 49 m/90 f 268 PD and nonH, mean age:69.6 y, 138 m/130 f 72/401 HC, mean age: 69.2 y, 187 m/214 f	Lifetime M: 9.6% 1-year M: 6.7%	UK PD Society Brain Bank	ICHD-II	Semi-structured headache questionnaire	Motor HY, MDS-UPDRSI, II, III Non-motor EDS, MMSE, PDSS-2, RBDSQ-J, PSQI	No effect of antiparkinsonian therapy on headaches	Lower lifetime M prevalence (9.6% vs. 18.0%) and 1 year (6.7% vs. 11.0%) PD vs. HC. Higher UPDRS III in PD and M vs. PD and nonM.
Heilbron, 2019 [27]	Case–control, *n* = 161,372 13,196 PD, mean age:69.8 y, 13,190 m/5126 f PD and M(ICHD): 8507 PD and MWA(ICHD): 8215 PD and MA (ICHD):8204 PD and M (broad Mdiagnosis): 11,052 PD and M:9532 C and M(ICHD): 144,619 C and MWA(ICHD): 139,655 C and MA (ICHD): 139,468 C and M (broad Mdiagnosis): 154,728 C and M: 162,044	PD and M(ICHD): 64.4% PD and MWA(ICHD): 62.2% PD and MA (ICHD): 62.1% PD and M (broad Mdiagnosis): 83.7% PD and M: 72.2%	NA	ICHD	NA	NA	NA	Association between PD and M remained significant (OR = 1.756 (1.554–1.985), *p* = 1.93 × 10^−19^) after correcting for head injury, angina and chest pain during exercise.
Sampaio Rocha Filho, 2020 [30]	Cross-sectional, *n*= 43, 12 PD and M, mean age:66 y, 24 m/22 f 31 PD and TTH, mean age:66 y, 24 m/22 f	M: 26% TTH: 67%	NA	ICHD-3 beta	Semi-structured questionnaire	Motor HY, MDS-UPDRSIII ON stage Non-motor ESS, HAD	NA	No association between headache and PD motor/non-motor symptoms.

AGES: age, gene/environment susceptibility; C: controls; CI: confidence interval; CURS: Columbia University rating scale; EDS: excessive daytime sleepiness; ESS: Epworth sleepiness scale; f: female; ICHD: International Classification of Headache Disorders; H: headache; HAD: Hospital Anxiety and Depression Scale; HC: healthy controls; HR: hazard ratio; HY: Hoehn and Yahr; HRS: Hamilton Rating Scale for Depression; m: male; M: migraine; MA: migraine with aura; MWA: migraine without aura; MMSE: mini–mental state examination; NA: not available; TTH: tension-type headache; TVH: tension–vascular headache; UPDRS: unified Parkinson’s disease rating scale; OR: odds ratio; PD: Parkinson’s disease; PDSS-2: PD sleep scale; RBDSQ-J: Rapid Eye Movement Sleep Behavior Disorder (RBD) Screening Questionnaire; PSQI: Pittsburgh Sleep Quality Index; STAI X-1: State–Trait Anxiety Inventory; WRS: Webster Rating Scale; and Zung RS: Zung self-rating scale * Headache prevalence is reported for PD patient.

## Data Availability

Not applicable.

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
