# Peer review of "Migraine, Tension-Type Headache and Parkinson’s Disease: A Systematic Review and Meta-Analysis"

_medicina, 2022, doi:10.3390/medicina58111684_

Round 1
Reviewer 1 Report
The manuscript reviews and analyses the relationship between migraine (M)/tension-type headache (TTH) and Parkinson's disease (PD).
This reviewer has major concerns regarding the design, hypothesis, and interpretation of the results in the discussion. First, the authors do not present any scientific grounds for why they hypothesize that patients with migraine or TTH have a higher risk for PD in later life. In contrast, M is known to be less problem after 50 years of age, especially for women. There is no discussion in the manuscript pointing out the effect of age on M och TTH. Furthermore, the discussion about possible mechanisms regarding dopaminergic/serotonergic systems in PD does not relate to any pathophysiological mechanisms known for migraine.
This reviewer highly suggests the authors reformulate the introduction, results, and discussion pointing out possible shared pathophysiological mechanisms between M and PD. However, according to this reviewer, it is difficult to find this relationship on the basis of current literature on pathophysiology.
Author Response
Thank you very much for reviewing our manuscript entitled " Migraine, tension-type headache and Parkinson's disease: A Systematic review and Meta-analysis " and for giving us the opportunity to publish it. We addressed all the issues raised by the reviewers.
We have amended the paper in line with the referees’ comments in yellow colour and the revised version is attached. We have dealt with the referees’ comments as follows:
Comments and Suggestions for Authors
Reviewer 1
The manuscript reviews and analyses the relationship between migraine (M)/tension-type headache (TTH) and Parkinson's disease (PD).
This reviewer has major concerns regarding the design, hypothesis, and interpretation of the results in the discussion. First, the authors do not present any scientific grounds for why they hypothesize that patients with migraine or TTH have a higher risk for PD in later life. In contrast, M is known to be less problem after 50 years of age, especially for women. There is no discussion in the manuscript pointing out the effect of age on M och TTH. Furthermore, the discussion about possible mechanisms regarding dopaminergic/serotonergic systems in PD does not relate to any pathophysiological mechanisms known for migraine.
This reviewer highly suggests the authors reformulate the introduction, results, and discussion pointing out possible shared pathophysiological mechanisms between M and PD. However, according to this reviewer, it is difficult to find this relationship on the basis of current literature on pathophysiology.
OUR RESPONSE We perform a systematic- review and meta-analysis according the most valided methodology. our search strategy had the advantage of being rigorous, allowing an exhaustive literature review. Other limitations consist of high heterogeneity in the estimation of the prevalence and ORs, the inclusion of studies with small sample size, and lack of meta-regression analysis. While the number of studies available for analysis was small and the majority of studies were cross-sectional and selection bias could not be ruled out, a strength of our study lies in the quality assessment of included studies. In particular, according to the NOS scores, the majority of the studies included had a score of ≥6, suggesting a moderate quality, which is one of the good points of this review.
We reformulate the Introduction: ‘’Primary headaches, migraine and tension type headache (TTH) are common pain disorders with comorbidities, such as cardiovascular, stroke, depression and dementias, and a substantial socioeconomic impact [7,8] Over the previous two decades, several studies have suggested a potential relationship between primary headaches and PD [9-11], while other studies have produced conflicting outcomes [12,13]. Barbanti et al. identified a lifetime migraine prevalence of 27.8% and a current migraine prevalence of 13.1% in a population of PD patients[14]. Moreover, a cohort study showed that subjects with a midlife history of headache and particularly those with migraine with aura had an increased likelihood of PD[9]. By contrast, Nunes et al. [15]demonstrated that PD patients had a lower lifetime prevalence of headache than controls. Previous studies have focused on the relationship between migraine and PD, and a prior validation study reported that 80% of individuals with non-migrainous headache might have TTH [16]. ‘’
The present meta- analysis did not demonstrate any effect of age on M or TTH . M is known to be less problem after 50 years of age, especially for women. However in the discussion part we have already pointed out as a limitation the fact that: ‘’ The majority of included studies of the present review was not designed to account for confounding factors. First, many women with PD in the 6th decade go through menopause and it is well-known that the hormonal transition during menopause has a significant association with migraine.’’ (lines page)
We discussed more in detail possible shared pathophysiological mechanisms between M and PD as well as between TTH and PD: “ Dopamine may be implicated in modulating pain processing in different areas of the brain, such as the spinal cord, thalamus, peri-aqueductal gray matter, basal ganglia, and cingulate gyrus. The mesolimbic, and nigrostriatal pathways originate in the substantia nigra pars compacta and project to the dorsal striatum [32] . These pathways are linked with various NMS in patients with PD, such as pain. Moreover, the pain tolerance threshold in patients with PD tends to be lower than that exhibited by healthy individuals. This could be ascribed to an damage of the descending inhibition of nociception from supraspinal structures[33]. Inadequate descending pain modulation has also been reported to be instrumental in the pathophysiology of chronic TTH [34]. Consequently, impaired dopaminergic function and deficient descending pain inhibition from supraspinal sites are emerge as potential shared mechanisms between PD and TTH. Serotonergic dysfunction has also been implicated in the pathophysiology of PD[35], with serotonergic cell loss having been described to appear even before nigrostriatal dopaminergic degeneration in PD [36]. In fact, previous animal studies demonstrate that serotonin release is blocked following the blockade of postsynaptic D2 receptors, and that serotonin release in the forebrain entails an intact local dopaminergic neurotransmission[37]. Furthermore, serotonergic deficiency assists in impaired descending inhibition of pain and central sensitization of nociceptive pathways, which are thought to be implicated in the pathophysiology of TTH [38]. In the light of the overlap between these processes, TTH may have a potential pathological link with PD.
Common mechanisms linking migraine with PD have also been suggested, namely dysfunction of the serotonergic and dopaminergic system. Serotonin depletion has been associated with increased cortical neuron sensitivity, vasoconstriction induced by cortical spreading depression in an animal model of migraine [39],[40]. Except from PD, dopaminergic dysfunction has also been hypothesized as a causal factor in migraine pathogenesis, as excessive yawning, nausea, and vomiting are common prodromal and accompanying symptoms of migraine. This could be related to stimulation of the DA2 receptors which are abundant in migraineurs with aura[41]. Iron, an essential nutrient for DA regulation, were elevated in the substantia nigra of PD patients, leading to toxic oxygen radicals and DA cell death [42]. Furthermore, increased iron accumulation in deep brain nuclei in migraineurs may be suggestive of impaired antinociceptive pathways [43]. So far, whether “Iron accumulation could be an indice of tissue damage due to other processes, or a reversible epiphenomenon’’ cannot be answered [42]. If the increased iron deposition observed in migraineurs in these few studies reflects (directly or indirectly) neuronal damage, this would be consistent with the late-life PD symptoms. By contrast, studies included in the present review did not investigate the relationship between iron accumulation and migraine. ‘’ (see in discussion part, page 5 lines123-164 )

Reviewer 2 Report
I believe there is little clarity in the conclusionsAuthor Response
Thank you very much for reviewing our manuscript entitled " Migraine, tension-type headache and Parkinson's disease: A Systematic review and Meta-analysis " and for giving us the opportunity to publish it. We addressed all the issues raised by the reviewers.
We have amended the paper in line with the referees’ comments in yellow colour and the revised version is attached. We have dealt with the referees’ comments as follows:
Comments and Suggestions for Authors
Reviewer 2
I believe there is little clarity in the conclusions
OUR RESPONSE: We clarify the conclusions in the abstract:’’ Observational data suggest that migraine and TTH could be linked to PD, but current literature is conflicting’’
We clarify the conclusions at the end of the manuscript: ‘’This systematic review revealed that migraine and TTH commonly co-exist with PD. However, the potentially higher risk of incident PD among migraineurs in cohort studieswas not verified (and even reversed) in other sensitivity analyses. Future research should aim to elucidate the abovementioned associations and their common mechanisms via large well-designed prospective multicenter studies.’’(page 7,lines 222-229)

Reviewer 3 Report
The primary concern for the authors to address is grammatical English. There are uncommon structures and some grammatical errors.
1. There are different colors and word formats in the manuscript.
2. Grammar corrections throughout the entire manuscript.
E.g., there are no spaces between words. For example, L19
The authors should try to shorter the structures to avoid misunderstandings
3. Please, rephrase this structure “but the relationship is complex.”
Author Response
Thank you very much for reviewing our manuscript entitled " Migraine, tension-type headache and Parkinson's disease: A Systematic review and Meta-analysis " and for giving us the opportunity to publish it. We addressed all the issues raised by the reviewers.
We have amended the paper in line with the referees’ comments in yellow colour and the revised version is attached. We have dealt with the referees’ comments as follows:
Comments and Suggestions for Authors
Reviewer 3
The primary concern for the authors to address is grammatical English. There are uncommon structures and some grammatical errors.
- There are different colors and word formats in the manuscript. OUR RESPONSE: We corrected them
- Grammar corrections throughout the entire manuscript. OUR RESPONSE: We corrected them
E.g., there are no spaces between words. For example, L19 OUR RESPONSE: We corrected them
The authors should try to shorter the structures to avoid misunderstandings
OUR RESPONSE: We corrected all grammar, word formats and space through the manuscript.
- Please, rephrase this structure “but the relationship is complex.” OUR RESPONSE: we rephrase it as suggested

Round 2
Reviewer 1 Report
Thanks for updating the manuscript
Reviewer 3 Report
1. There are still essential concerns regarding grammatical language. It is advised professional editing service. E.g., punctuation and parallelism.